# Risk of Infertility in Males with Obstructive Sleep Apnea: A Nationwide, Population-Based, Nested Case-Control Study

**DOI:** 10.3390/jpm12060933

**Published:** 2022-06-05

**Authors:** Pin-Yao Lin, Hua Ting, Yen-Ting Lu, Jing-Yang Huang, Tsung-Hsien Lee, Maw-Sheng Lee, James Cheng-Chung Wei

**Affiliations:** 1Institute of Medicine, Chung Shan Medical University, Taichung 406, Taiwan; huating101@gmail.com (H.T.); ytlu.tw@gmail.com (Y.-T.L.); wchinyang@gmail.com (J.-Y.H.); jackth.lee@gmail.com (T.-H.L.); msleephd@gmail.com (M.-S.L.); 2Division of Infertility, Lee Women’s Hospital, Taichung 406, Taiwan; 3Sleep Medicine Center, Chung Shan Medical University Hospital, Taichung 402, Taiwan; 4Department of Physical Medicine and Rehabilitation, Chung Shan Medical University Hospital, Taichung 402, Taiwan; 5Department of Otolaryngology, St. Martin De Porres Hospital, Chiayi 600, Taiwan; 6Department of Otolaryngology, Chung Shan Medical University Hospital, Taichung 402, Taiwan; 7Department of Medical Research, Chung Shan Medical University Hospital, Taichung 402, Taiwan; 8Department of Obstetrics and Gynecology, Chung Shan Medical University Hospital, Taichung 402, Taiwan; 9Department of Internal Medicine, Chung Shan Medical University Hospital, Taichung 402, Taiwan; 10Graduate Institute of Integrated Medicine, China Medical University, Taichung 402, Taiwan

**Keywords:** sleep apnea, obstructive sleep apnea (OSA), National Health Insurance Research Database (NHIRD), hypoxia

## Abstract

Obstructive sleep apnea (OSA) yields intermittent hypoxia, hypercapnia, and sleep fragmentation. OSA is associated with chronic medical conditions such as cardiovascular diseases, metabolic syndrome, and neurocognitive dysfunction. However, the risk of infertility in OSA remains unclear due to limited data and lack of long-term population-based studies. The study aims to assess the risk of infertility in obstructive sleep apnea (OSA) by means of a population-based cohort study. The data was utilized from the Taiwan National Health Insurance Research Database (NHIRD) to conduct a population-based cohort study (1997–2013). Compared with the Non-OSA group, the male with OSA and surgery group has the OR (odds ratio) of infertility of 2.70 (95% CI, 1.46–4.98, *p* = 0.0015), but no significance exists in females with OSA. When the data was stratified according to age and gender, some associations in the specific subgroups were significant. Respectively, males aged 20–35 years old and aged 35–50 years old with a history of OSA and surgery both had a positive association with infertility. (aOR: 3.19; 95% CI, 1.18–8.66, *p* = 0.0227; aOR: 2.57; 95% CI, 1.18–5.62 *p* = 0.0176). Male patients with OSA suffer from reduced fertility, but no significant difference was noted in females with OSA. The identification of OSA as a risk factor for male infertility will aid clinicians to optimize long-term medical care. Furthermore, more studies will be encouraged to clarify the effect of OSA on female fertility.

## 1. Introduction

Infertility, defined as the failure to achieve a clinical pregnancy after 12 months of regular unprotected sexual intercourse, affects 8–12% of reproductive-aged couples worldwide. [1] Although infertility could potentially be attributed to several causes such as male factors, tubal patency, uterine factors, and hormonal function, “unexplained infertility” is frequently encountered in cases with normal ovulatory function along with tubal patency and a normal semen analysis.

Characterized by recurring complete or incomplete collapse of the upper airway during sleep, obstructive sleep apnea (OSA) yields intermittent hypoxia, hypercapnia, and sleep fragmentation [2]. OSA is a highly prevalent disorder, affecting ~20% of Americans and an even higher proportion of Asians [3]. Importantly, OSA is associated with a number of chronic medical conditions such as cardiovascular diseases, metabolic syndrome, and neurocognitive dysfunction [4,5,6].

Potential associations between OSA and the likelihood and outcome of infertility have recently emerged, suggesting that OSA might cause male infertility or reduce fertility [7,8,9,10,11]. Torres et al. [12] discovered that intermittent hypoxia significantly increased testicular oxidative stress and consequently reduced progressive sperm motility in mice. However, longitudinal studies examining the relationship between OSA and human infertility are lacking. Therefore, the current study examined whether sleep apnea is a risk factor for incidental infertility in females and males.

## 2. Materials and Methods

### 2.1. Data Sources

Since 1995, Taiwan has established a compulsory National Health Insurance (NHI) program, which currently covers 99.5% of the 23 million Taiwanese via 97% of the clinics enrolled in this system. This nested case-control study used data obtained from the Longitudinal Health Insurance Database 2000 (LHID 2000) spanning the period 1997–2013. The LHID 2000 is a sub-dataset of all information—socio-demographic status, outpatient, inpatient, and emergency care, surgical treatment, and prescription drugs—collected from one million randomly selected representatives by the National Health Insurance Research Database (NHIRD). For privacy concerns, all personal identification data in the NHIRD were anonymized and de-identified. The coding of clinical diagnoses in the NHIRD was done according to the International Classification of Diseases, Ninth Revision, Clinical Modification (ICD-9-CM). Our study protocol was approved by the Institutional Review Board of St. Martin De Porres Hospital, Chiayi, Taiwan. (IRB: 20B-013, 12 June 2020).

### 2.2. Study Participants

Patients diagnosed with new-onset infertility (ICD-9-CM: 628.0-9, 606.0–606.9) between January 1997 and December 2013 were categorized as the case group (*n* = 14,997). To improve the validity, the diagnosis of OSA was defined on the basis of at least three outpatient visits or one admission. The date of the first diagnosis as infertility was defined as the index date. The exclusion criteria were as follows: (1) Index date of Infertility was before 2000 (*n* = 1932); (2) Age < 20 years old and >50 years (*n* = 324). Finally, 12,741 (male, *n* = 1623; female, *n* = 11,118) infertility cases and four-fold as many age- and sex-matched non-infertility controls (total *n* = 50,964: male, *n* = 6492; female, *n* = 44,472) were included in this study. We performed propensity score matching (PSM) at a 1:2 ratio to minimize the potential confounding effects of sex, age, and selected comorbidities on the incidence of infertility to make the frequency of selected variables uniform in the study and control cohorts. The matched control groups were retrieved during the same period. Then, we included PSM to identify 12,718 cases (male, *n* = 1607; female, *n* = 11,111) with infertility and 25,436 individuals (male, *n* = 3,214; female, *n* = 22,222) without infertility. Figure 1 illustrates our study design.

### 2.3. Outcomes and Relevant Variables

Patients diagnosed with OSA (ICD-9-CM code 780.51, 780.53, 780.57) with at least three outpatient visits or one hospital admission before the index date were defined as level I outcomes for this study. Patients diagnosed with OSA and who underwent surgery, such as uvulopalatopharyngoplasty (UPPP) or septomeatal plasty, were defined as level II outcomes. The following common comorbidities were listed to compare the baseline characteristics between both groups and evaluate the confounding effects: overweight and obesity (ICD-9-CM code: 278), type 2 DM (ICD-9-CM code: 250), hyperlipidemia (ICD-9-CM code: 272.0–270.2), chronic obstructive pulmonary disease (COPD; ICD-9-CM codes: 490–496), chronic kidney disease (ICD-9-CM code: 585), chronic liver diseases (ICD-9-CM code: 571 and 573), major psychiatric disorders (ICD-9-CM code: 290–298), and minor psychiatric disorders (ICD-9-CM code: 300–302, 306–311, 316). Both the study and control groups were followed till the end of 31 December 2013. For women, the number of pregnancies between January 1997 and the index date was also calculated.

### 2.4. Statistical Analysis

The absolute standardized difference (ASD) was used to evaluate the balance of the baseline characteristics in the age- and sex-matched and propensity score-matched populations. Generally, ASD < 0.1 is considered a small difference. Conditional logistic regression was used to estimate the adjusted odds ratio (aOR) and 95% confidence interval (CI) for the risk of infertility. Statistical significance was defined as *p* < 0.05. All statistical analyses were performed using SAS version 9.4 software (SAS Institute, Cary, NC, USA). In addition, conditional logistic regression analyses were performed to examine the risk of comorbidities for these two groups, after adjusting for sociodemographic characteristics including geographic region, monthly income, and urbanization level.

## 3. Results

### 3.1. Baseline Characteristics of the Study Population

The eligible study participants comprised 12,741 patients in the infertility group and 50,964 age- and sex-matched individuals in the control (comparison) group. Additionally, for comparison, PSM was conducted at a ratio of 1:2. Finally, 12,718 patients with infertility and 25,716 controls after PSM (Figure 1) were included. The prevalence of comorbidities in the age- and sex-matched population is shown in Table 1. At baseline, the frequencies of the selected factors, including age, sex, monthly income, urbanization, year of cohort entry, and co-morbidities, were averaged equally in each cohort. Following PSM, most baseline characteristics were balanced. Women in the infertility group had a higher rate of non-conception during the period from January 1997 to their index date.

### 3.2. Odds Ratio of Infertility with OSA and without OSA

Table 2 shows the results of conditional logistic regression for the estimation of the odds ratio of infertility in age- and sex-matched groups and the presence of comorbidities. Compared with the Non-OSA group, the male OSA with surgery group had a significantly different aOR of infertility of 2.70 (95% CI, 1.46–4.98, *p* = 0.0015); however, no significant difference was observed in the female OSA with surgery group. The aORs of various comorbidities for subjects with and without infertility are shown in Table 2. After adjusting for variables, a conditional logistic regression was performed to compare the risk of each comorbidity between the infertility and non-infertility groups. Females with infertility were likely to have longer hospital stays, COPD, chronic liver disease, chronic kidney disease, and major/minor psychiatric disorders (*p* < 0.05; Table 2). Males with infertility were likely to have COPD, chronic liver disease, chronic kidney disease, and major/minor psychiatric disorders (*p* < 0.05; Table 2). Table 3 shows the results of the conditional logistic regression for the estimation of odds ratios of infertility in the propensity-score-matched groups. When we stratified the data according to age and gender, we found some significant associations in specific subgroups. For instance, a positive association between incidental infertility and therapeutic surgery for OSA in males aged 20–35 years and 35–55 years was observed (aOR: 3.19; 95% CI, 1.18–8.66, *p* = 0.0227; aOR: 2.57; 95% CI, 1.18–5.62 *p* = 0.0176, respectively) (Table 4).

## 4. Discussion

OSA has been widely recognized not only as an important cause of cardiovascular morbidity and mortality, metabolic and cognitive dysfunction, and even cancer but also as a cause of impaired work performance and reduced quality of life [5,13]. The results of this retrospective, nationwide, population-based cohort study suggest that OSA is a significant risk factor for infertility only in males, particularly in males with OSA and OSA surgery (2.7 times higher risk of infertility compared with the control population). There have been few longitudinal studies examining the association between OSA and human infertility. Our results are consistent with those of previous studies and support their findings via this population-based long-term follow-up.

A recent systematic review and meta-regression analysis report a significant decline in sperm counts over the past four decades, a trend that is ongoing [14,15]. This deterioration in male reproductive health has encouraged more researchers to consider the effects of environmental and lifestyle factors on male reproductive potential. Although the mechanisms underlying OSA’s effects on male infertility are beyond the scope of this study and are not completely clear, the observed correlation between OSA and male infertility could be explained from several aspects. First, previous studies have revealed the correlation of OSA and the deviations in reproductive hormone profiles such as testosterone and prolactin. While significantly lower serum testosterone levels were found in the OSA group [11,16,17,18], the effectiveness of OSA therapy in restoring reproductive hormone levels is still controversial [19,20,21,22], possibly suggesting that the association between the severity of OSA and low testosterone levels is unclear. Kouchiyama et al. found that more severe nocturnal oxygen desaturation in OSA was associated with a delayed peak testosterone level, and that the total desaturation time was negatively correlated with serum testosterone levels. [23] These authors proposed that nocturnal hypoxemia could contribute to an aberration in the hypothalamic-pituitary gonadal axis as a potential cause of infertility. Other study findings have suggested that sleep fragmentation was associated with a diminished nocturnal rise in serum testosterone level as a possible explanation of the lower testosterone levels observed in OSA [24,25]. Another proposed hormone linking OSA and infertility is the abnormal secretion of prolactin (PRL) after sleep fragmentation and hypoxia [7,26]. PRL receptors are present in the male reproduction organs; however, the precise function of this hormone in male reproductive physiology remains unspecified. PRL is believed to control gonadal function indirectly by controlling the secretion of gonadotropin-releasing hormone/gonadotropins from the hypothalamus/pituitary or by directly increasing the concentration of luteinizing-hormone receptors on Leydig cells in the testes [27]. Therefore, PRL may be a cause for infertility in OSA. Secondly, studies have shown marked generation of reactive oxygen species (ROS) from leukocytes, reduced plasma levels of nitrite and nitrate, increased lipid peroxidation, and reduced antioxidant capacity in OSA patients [28]. High ROS levels not only induce lipid peroxidation and disrupt DNA and RNA but also impair protein functions in the spermatozoa and other testicular cells. ROS also adversely affect male reproductive functions and may induce infertility indirectly by affecting the hypothalamus-pituitary-gonadal (HPG) axis and/or disrupting its crosstalk with other hormonal axes [29,30]. Third, OSA is associated with male sexual function disorder such as erectile dysfunction, ejaculation disorder, and decreased sexual frequency [31,32,33]. Impaired sexual function is also possibly a contributing factor toward infertility.

Our study also found that OSA with surgery significantly increased the risk of male infertility. Continuous positive airway pressure (CPAP) is considered the gold standard for OSA treatment and is highly efficacious in controlling OSA symptoms [6,34]. However, surgery may be reserved for higher severity of OSA or CPAP-refractory cases. Thus, the severity of OSA may be positively correlated with male infertility. However, higher severity of OSA is associated with other comorbidities such as physical activity, smoking habits, alcohol consumption, diet, family history, and body mass index that are also contributing factors to male infertility.

Another noteworthy finding from our study is that OSA did not affect female infertility. OSA is a severe public health disease affecting 3–7% of men and 2–5% of women [35], and there are significant gender differences in the pathological mechanism, symptoms presentation, and disease manifestation in the link between OSA and dys-metabolism [36]. Gender differences regarding OSA might possibly explain why OSA affects male infertility instead of female infertility in consequence. Previous studies demonstrated the strong association between sleep disorder and female infertility [9,37,38]. However, to date, there is little evidence of the correlation between OSA and female infertility. The possible reason is that infertility may arise from male factors, female factors, or a combination of these. The sole male factor is only responsible for 26–30% of sub-fertile couple, the majority influenced by female or combined factors [39]. Due to the more complicated etiologies of female infertility, OSA might not have significant impact on female infertility. Another possible explanation is that the female reproductive system seems more adaptive to hypoxia status than the male reproductive system. A study [40] examining in vivo oxygen levels in the female reproductive tract and their importance in human conception found that physiological oxygen concentration in the female reproductive tract is lower, i.e., 2–8% [41], and that embryo development is improved when cultured in lower oxygen concentrations of 5–7% compared to 20% in other mammals. Bontekoe et al. showed that culturing in a constant level of low-oxygen concentration may improve the success rates of in vitro fertilization and intracytoplasmic sperm injection and increase live birth rates [42]. 

Jhuang et al [43] conducted a large case-control study revealing that infertile men had a 1.24 fold higher risk of OSA compared with control group. The study defines OSA as primary outcome to find if infertile men have an association with OSA. However, our study hypothesized that OSA was an exposure to risk, then looked to see if infertility was the result. Although different methods were used in these two studies, the conclusion was similar, supporting that OSA is associated with and a possible cause of infertility. Another retrospective cohort study by Lim et al. [44] found that OSA is associated with female infertility. Some discrepancies exist between these two studies for the following reasons: firstly, the causes of female infertility are far more complicated than those of male infertility. Age is the most decisive factor and the mean age in this study was 32 years old. The study population was relatively too young to recruit affected infertile women. Secondly, prevalence data show that more men than women are affected by OSA. OSA has been estimated to have a male-to-female ratio of between 3 to 1 and 5 to 1 in the general population and an even higher ratio of between 8 to 1 and 10 to 1 in some clinical groups [45]. The study indicated that, in women with infertility, only 1.38% had a history of OSA compared with 0.63% of fertile controls. The low incidence rate of OSA compared with the general female population [36] may be explained by the fact that OSA is more common in older women than in younger. Nevertheless, the limitation in the number of participants might possibly conclude that further study is needed to better clarify the association of OSA with female infertility. Indeed, difficulties remain in studying such an association, such as the more diverse causes of female infertility, the prevalence of OSA in women, and the lag time from OSA to possible manifestation as infertility.

Our study has several strengths. First, this was a nationwide study on the association between OSA and the subsequent development of male infertility. Although the prevalence and incidence rates of OSA in infertile patients are difficult to determine due to the lack of definite screening and are as yet unknown, by retrospectively analyzing infertile patients and their counterparts our study shows that incidental infertility in males significantly correlated with the combination of past history of OSA with surgical intervention as a therapeutic strategy. Second, being based on a nationwide data collection, this study provided a sufficiently large sample size to avoid selection bias. Third, the long-term follow-up of up to 16 years renders the results more reliable. However, certain limitations still exist. First, the cause–effect association was not established and the study only concluded an association between OSA and male infertility. More investigations on pathophysiology and mechanism are necessary in the future. Second, the study included patients with OSA from the LHID 2000 database. Thus, patients who may not have been adequately diagnosed with OSA were included in the cohort, and the prevalence/incidence of OSA in infertile patients could not be identified in the study. Furthermore, another limitation is rooted in the lack of detailed anthropometric measures such as physical activity, smoking habits, alcohol consumption, diet, family history, and body mass index that are not provided by Taiwan NHIRD, despite these factors potentially confounding the effect of OSA on incidental infertility. Additionally, imaging results, pathologic findings, and biochemical data are not available in the NHIRD. Further prospective observational and interventional studies with more lifestyle information are necessary to support the findings. The subgroup analysis for types of male infertility was not analyzed, because precise categorizing of these subgroups in clinical practice is difficult. Future study to analyze azoospermia and non-testicular causes may enhance the clinical significance.

## 5. Conclusions

Our study of a nationwide, population-based, long-term follow-up demonstrates that males with OSA are likely to suffer from infertility. The identification of OSA as a risk factor of male infertility would aid clinicians in optimizing medical care. Fertility counseling and interventions should be suggested to men with OSA along with exploration of whether effective treatment for OSA would ameliorate reduced male fertility. Furthermore, more studies are encouraged to clarify the effect of OSA on female fertility.

## Figures and Tables

**Figure 1 jpm-12-00933-f001:**
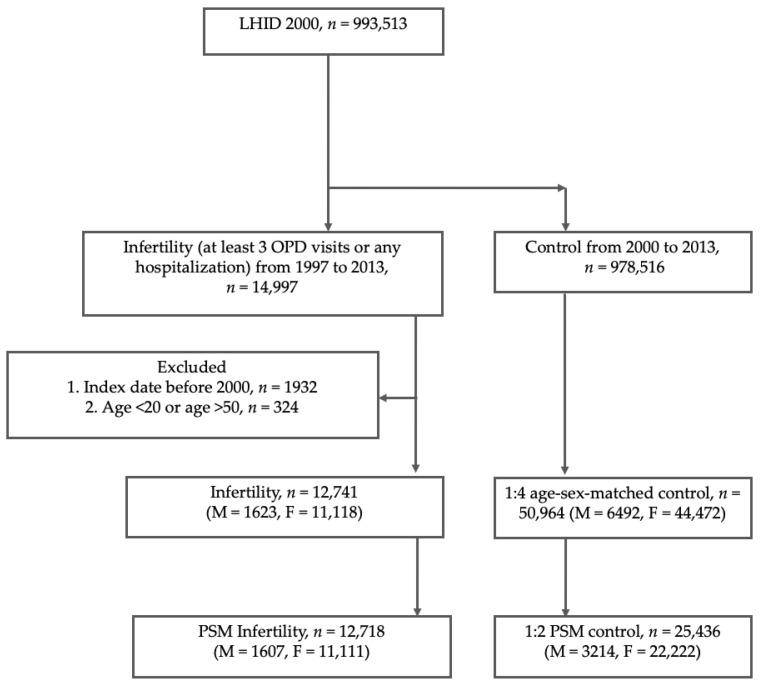
Study Design. Abbreviations: LHID = Longitudinal Health Insurance Database; OPD = outpatient department; M = male; F = female; PSM = propensity score matching.

**Table 1 jpm-12-00933-t001:** Baseline characteristics among age- and sex-matched population.

	Female		ASD	Male		ASD
	Control	Infertility		Control	Infertility	
	*n* = 44,472	*n* = 11,118		*n* = 6492	*n* = 1623	
Study group			0.00			0.14
Non-OSA	44,169 (99.32%)	11,029 (99.20%)		6428 (99.01%)	1598 (98.46%)	
OSA without surgery	264 (0.59%)	77 (0.69%)		37 (0.57%)	7 (0.43%)	
OSA with surgery	39 (0.09%)	12 (0.11%)		27 (0.42%)	18 (1.11%)	
Age at index date			0.00			0.00
20 to <25	4200 (9.44%)	1050 (9.44%)		84 (1.29%)	21 (1.29%)	
25 to <30	13,140 (29.55%)	3285 (29.55%)		760 (11.71%)	190 (11.71%)	
30 to <35	17,000 (38.23%)	4250 (38.23%)		2632 (40.54%)	658 (40.54%)	
35 to <40	7244 (16.29%)	1811 (16.29%)		1928 (29.70%)	482 (29.70%)	
40 to <45	2264 (5.09%)	566 (5.09%)		764 (11.77%)	191 (11.77%)	
45 to 55	624 (1.40%)	156 (1.40%)		324 (4.99%)	81 (4.99%)	
Urbanization			0.07			0.04
Urban	28,490 (64.06%)	7220 (64.94%)		4002 (61.65%)	1029 (63.40%)	
Sub-urban	12,560 (28.24%)	3057 (27.50%)		1951 (30.05%)	482 (29.7%)	
Rural	3422 (7.69%)	841 (7.56%)		539 (8.30%)	112 (6.90%)	
Low income	160 (0.36%)	45 (0.40%)	0.01	29 (0.45%)	3 (0.18%)	0.05
Length of hospital stays			0.24			0.12
0 day	36,945 (83.07%)	9494 (85.39%)		5947 (91.61%)	1481 (91.25%)	
1–6 days	6041 (13.58%)	1352 (12.16%)		350 (5.39%)	95 (5.85%)	
≥7 days	1486 (3.34%)	272 (2.45%)		195 (3.00%)	47 (2.90%)	
Co-morbidity						
Overweight and obesity	222 (0.50%)	71 (0.64%)	0.02	21 (0.32%)	6 (0.37%)	0.01
Diabetes mellitus	825 (1.86%)	173 (1.56%)	0.02	158 (2.43%)	55 (3.39%)	0.06
Hyperlipidemia	394 (0.89%)	114 (1.03%)	0.01	150 (2.31%)	59 (3.64%)	0.08
Asthma	1288 (2.90%)	306 (2.75%)	0.01	131 (2.02%)	36 (2.22%)	0.01
COPD	1848 (4.16%)	530 (4.77%)	0.03	239 (3.68%)	95 (5.85%)	0.10
Chronic liver diseases	410 (0.92%)	143 (1.29%)	0.03	92 (1.42%)	45 (2.77%)	0.09
Chronic kidney diseases	2534 (5.70%)	859 (7.73%)	0.08	665 (10.24%)	267 (16.45%)	0.18
Major psychiatric disorders	718 (1.61%)	257 (2.31%)	0.05	120 (1.85%)	49 (3.02%)	0.08
Minor psychiatric disorders	3351 (7.54%)	1105 (9.94%)	0.09	474 (7.30%)	163 (10.04%)	0.10
Pregnancy between 1997 and index date			1.32			
Never pregnant	29,684 (66.75%)	9182 (82.59%)				
1 time	8961 (20.15%)	1587 (14.27%)				
2 times	4820 (10.84%)	304 (2.73%)				
3 times	862 (1.94%)	37 (0.33%)				
>3 times	145 (0.33%)	8 (0.07%)				

Note: Values are *n* (%); OSA = obstructive sleep apnea; COPD = chronic obstructive pulmonary disease; ASD = absolute standardized difference was used to evaluate the balance of the baseline characteristics in the age- and sex-matched and propensity score-matched populations.

**Table 2 jpm-12-00933-t002:** Conditional logistic regression for estimation of odds ratio of Infertility in age- and sex-matched groups.

Variable	Female		Male	
	aOR (95% C.I.)	*p*-Value	aOR (95% C.I.)	*p*-Value
Study group				
Non-OSA	Reference		Reference	
OSA without surgery	1.15 (0.89–1.49)	0.2881	0.72 (0.32–1.64)	0.4360
OSA with surgery	1.17 (0.61–2.24)	0.6388	2.70 (1.46–4.98)	0.0015 *
Urbanization				
Urban	Reference		Reference	
Sub-urban	0.96 (0.91–1.00)	0.0739	0.97 (0.86–1.09)	0.6001
Rural	0.97 (0.89–1.05)	0.4313	0.81 (0.65–1.01)	0.0570
Low income	1.13 (0.81–1.57)	0.4886	0.36 (0.11–1.23)	0.1033
Length of hospital stays				
0 day	Reference		Reference	
1–6 days	0.87 (0.81–0.92)	<0.0001	0.99 (0.78–1.26)	0.9268
≥7 days	0.68 (0.59–0.77)	<0.0001	0.80 (0.57–1.12)	0.1894
Co-morbidity				
Overweight and obesity	1.23 (0.94–1.61)	0.1372	0.94 (0.37–2.38)	0.8994
Hyperlipidemia	1.06 (0.85–1.32)	0.6013	1.18 (0.85–1.64)	0.3280
Asthma	0.91 (0.80–1.03)	0.1481	0.96 (0.65–1.41)	0.8271
COPD	1.12 (1.02–1.24)	0.0226	1.48 (1.16–1.91)	0.0021
Chronic liver diseases	1.36 (1.12–1.65)	0.0021	1.73 (1.19–2.52)	0.0043
Chronic kidney diseases	1.34 (1.24–1.46)	<0.0001	1.58 (1.34–1.86)	<0.0001
Major psychiatric disorders	1.28 (1.11–1.49)	0.0011	1.40 (0.98–2.00)	0.0673
Minor psychiatric disorders	1.29 (1.19–1.38)	<0.0001	1.21 (0.99–1.47)	0.0625

Note: Data presented as aOR (95% C.I.). *p* < 0.05 was considered statistically significant and statistically significant values are marked with *. OSA = obstructive sleep apnea; COPD = chronic obstructive pulmonary disease; C.I. = confidence interval; aOR = adjusted odds ratio adjusted for urbanization, low income, length of hospital stays, and co-morbidity at index date.

**Table 3 jpm-12-00933-t003:** Conditional logistic regression for estimation of odds ratio for infertility in propensity-score-matched groups.

Variable	Female		Male	
	aOR (95% C.I.)	*p*-Value	aOR (95% C.I.)	*p*-Value
Study group				
Non-OSA	Reference		Reference	
OSA without surgery	1.23 (0.92–1.63)	0.1622	0.66 (0.26–1.68)	0.3797
OSA with surgery	0.83 (0.42–1.62)	0.5814	2.57 (1.28–5.17)	0.0080 *

Note: Data presented as aOR (95% C.I.). *p* < 0.05 was considered statistically significant and statistically significant values are marked with *. OSA = obstructive sleep apnea; C.I. = confidence interval; aOR = adjusted odds ratio adjusted for urbanization, low income, length of hospital stays, and co-morbidity at index date.

**Table 4 jpm-12-00933-t004:** Subgroup analysis for the odds ratio (95% C.I.) stratified by age groups (age- and sex-matched population).

Variable	Female		Male	
	aOR (95% C.I.)	*p*-Value	aOR (95% C.I.)	*p*-Value
20 to 35				
Study group				
Non-OSA	Reference		Reference	
OSA without surgery	1.09 (0.78–1.52)	0.6116	0.48 (0.11–2.11)	0.3332
OSA with surgery	0.96 (0.41–2.20)	0.9146	3.19 (1.18–8.66)	0.0227 *
35 to 55				
Study group				
Non-OSA	Reference		Reference	
OSA without surgery	1.27 (0.84–1.92)	0.2635	0.89 (0.32–2.49)	0.8300
OSA with surgery	1.57 (0.54–4.52)	0.4071	2.57 (1.18–5.62)	0.0176 *

Note: Data presented as aOR (95% C.I.). *p* < 0.05 was considered statistically significant and statistically significant values are marked with *. OSA = obstructive sleep apnea; C.I. = confidence interval; aOR = adjusted odds ratio adjusted for urbanization, low income, length of hospital stays, and co-morbidity at index date.

## Data Availability

All data is from the National Health Insurance Research Database (NHIRD) published by Taiwan National Health Insurance (NHI) Bureau. Due to legal restrictions imposed by the government of Taiwan concerning the “Personal Information Protection Act”, data is not publicly available.

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
