# Peer review of "Risk of Infertility in Males with Obstructive Sleep Apnea: A Nationwide, Population-Based, Nested Case-Control Study"

_jpm, 2022, doi:10.3390/jpm12060933_

Round 1

Reviewer 1 Report

This article dealt with a prevalent disorder (OSA) and its possible association with infertility. Due to the prevalence of OSA the study of the subject is important.

Excluding patients with  azoospermia and infertility with extratesticular causes could have added to the clinical significance of the findings 

"Our study also found that OSA with surgery significantly increased the risk of male infertility". The finding is an association which could also be explained by other comorbidities (physical activity, smoking habits, alcohol consumption, diet, family history, and body mass index  as discussed in the limitations)

To our knowledge, there have been no longitudinal studies examining the relationship between OSA and human infertility

Please see: Association of Obstructive Sleep Apnea With the Risk of Male Infertility in Taiwan (JAMA Netw Open. 2021;4(1):e2031846. doi:10.1001/jamanetworkopen.2020.31846)  - a similar study has been published

With relation to female infertility - please discuss the findings from "Obstructive sleep apnea increases risk of female infertility: A 14-year nationwide population-based study" Plos one 2021 Dec 15;16(12):e0260842.

 doi: 10.1371/journal.pone.0260842. eCollection 2021

What could be the reason for the contradicting findings.

Author Response

1. Thank you for the comments and concern.

  • The study aims to find if OSA is a factor for male infertility using nested case-control model. The subgroup analysis for types of male infertility was not analyzed yet for the reasons as following: (1) Although 40% of all patients showed primary causes of infertility, which could be subdivided based on the severity of their effect, but up to 75% of oligozoospermia cases remained idiopathic so far. [1] To precisely categorize these subgroups in clinical practice somehow is difficult. (2) While the diagnosis of azoospermia is rare (approximately 1% of all men), approximately 5%–15% of all infertile men. The anticipated contributing effect is low.[2] However, we total agree that future study to analyze the azoospermia, non-testicular causes may enhance the clinical significance. We emphasized this point of study’s entity and address the importance into the limitation. (line 259-262)

2. Thank you for the comment.

  • Indeed, our findings high lightened if OSA is so severe and intervention(surgery) is necessary, the male patient is at higher risk to get subfertility in the future. The higher severity of OSA highly associated with other comorbidities that are also contributing factors for male infertility. We added the explanation into discussion. (line 224-227)

3. Thank you for the comments and concern.

  • We are sorry not to update the latest publication on the issue. (Actually, the manuscript had been finished one year ago with delayed submission.) We deleted the improper sentences “there have been no longitudinal studies examining the relationship between OSA and human infertility” and changed to some similar studies…(line 182-183)
  •  We throughout read the papers listed from your comments and had some comparisons as followings:
    • Association of Obstructive Sleep Apnea With the Risk of Male Infertility in Taiwan (JAMA Netw Open. 2021;4(1):e2031846.)It is a very informative study. The large case-control study revealed infertile men had a 1.24-fold higher risk of OSA compared with control group, OSA is a risk factor for infertility. The study defines OSA as primary outcome to find if infertility men have association with OSA. However, our study design is different from these points. We hypothesized OSA was an “exposure” of risk then looked for if infertility was the “result”. So, we recruited infertile men and retrospective analyzed if they had the OSA before and its duration. Although different ways were used in these two studies, the conclusion was similar to support that OSA is associated and possible cause of infertility.
    • Obstructive sleep apnea increases risk of female infertility: A 14-year nationwide population-based study" Plos one 2021 Dec 15;16(12):e0260842.The study is also very informative on OSA and female infertility. Some discrepancies exist for the following reasons: Firstly, the causes of female infertility far more complicated than male infertility. Age is a most influential factor and the mean ages in this study were 32 y/o. The study population was relative too young to recruit affected infertile women. Secondly, OSA has historically been regarded as a male disease. OSA has been estimated to have a male-to-female ratio of between 3 : 1 and 5 : 1 in the general population and a much higher ratio of between 8 : 1 and 10 : 1 in some clinical groups. Prevalence data do show that more men than women are affected by OSA.[3] The study indicated women with infertility, only 1.38% had a history of OSA compared with 0.63% of fertile controls . The incidence rate is very low compared the data from the general female population (3-7%) [4] that may be explained OSA is more common in older women than young one. Lastly, although the study draws subjects from a large database, however, the number of infertility and OSA was only 33 and the number of OSA with no infertility in the control group was only 30 subjects. The cases is so small may be another bias for the results.

Sincerely, thank you for your nice suggestions again and let us revised the topic. We think it is important and addressed these into discussion. (line 248-264)

Reference:

1.         Punab, M.; Poolamets, O.; Paju, P.; Vihljajev, V.; Pomm, K.; Ladva, R.; Korrovits, P.; Laan, M. Causes of male infertility: a 9-year prospective monocentre study on 1737 patients with reduced total sperm counts. Hum Reprod 2017, 32, 18-31, doi:10.1093/humrep/dew284.

2.         Practice Committee of the American Society for Reproductive Medicine in collaboration with the Society for Male, R.; Urology. Evaluation of the azoospermic male: a committee opinion. Fertil Steril 2018, 109, 777-782, doi:10.1016/j.fertnstert.2018.01.043.

3.         Wimms, A.; Woehrle, H.; Ketheeswaran, S.; Ramanan, D.; Armitstead, J. Obstructive Sleep Apnea in Women: Specific Issues and Interventions. Biomed Res Int 2016, 2016, 1764837, doi:10.1155/2016/1764837.

4.         Martins, F.O.; Conde, S.V. Gender Differences in the Context of Obstructive Sleep Apnea and Metabolic Diseases. Front Physiol 2021, 12, 792633, doi:10.3389/fphys.2021.792633.

Reviewer 2 Report

Risk of Infertility in Males with Obstructive Sleep Apnea: A Nationwide, Population-based, Nested Case‒Control Study (Manuscript ID: jpm-1668278)

The current study assessed the risk of infertility, specifically, male infertility, in patients with obstructive sleep apnea using the data retrieved from a population-based cohort. Findings from this study showed that male patients with OSA between the ages of 20-35 years and 35-55 years have an Odds Ratio of 3.19 and 2.57 respectively when compared to the control group. This is an interesting outcome, but several questions arise when it comes to the study design and result presentation.

Herewith are my comments:

Specific comments

The concluding statement in the abstract section should be rephrased to represent the results of the study accurately. It presently reads “The identification of OSA as a risk factor for infertility would aid clinicians to optimize long-term medical care”. Since no significant differences were seen in the outcomes measured in the female cohort of the study, the conclusion should rather be focused on male infertility rather than both. Alternatively, a statement encouraging future studies to investigate the effect of OSA on female fertility can be included.

This study is more of a retrospective case-control study rather than a nested case-control study. A case-control study done in the population of an ongoing cohort study, is called "nested" inside the cohort. The current study retrieved data retrospectively. Please revise appropriately.

The abstract section reads “We utilized the data from Taiwan National Health Insurance Research Database (NHIRD) to 25 conduct a population-based cohort study (2000-2013)”, and the methods section reads “This nested case‒control study used data obtained from the Longitudinal Health Insurance Database 2000 (LHID 2000) spanning the period 1997–2013”. Which is which? Please clarify.

The abstract result section reads “Respectively, male aged 20-35 years old and aged 35-55 years old with history with OSA and surgery both 30 had positive association with infertility”, while it was stated in the methods that “the exclusion criteria were as follows: (1) Index date of Infertility was before 2000 (n = 1932); (2) Age < 20 years old and >50 years (n = 324)”. If > 50 years were excluded, where does the 55 years bracket in the abstract and results come from?

Authors should clearly state which variables were collected during a certain period. If the data that were retrieved for infertility starts in 2000, does this coincide with the duration for OSA?. That is, the index date for OSA should also be given.

Although it was briefly shown in Figure 1, it was not stated if the data for the control groups were retrieved during the same period in the method section.

Line 100- “For women, we also calculated the number of pregnancies between January 1997 and the index date”. Is the index date you are referring to 2000?

Do the authors think that the differences seen in the male outcome were because of the lower population size when compared to the female population. How was this adjusted? PSM was performed to adjust for inter-variable biase, how about the intra-variables?

Table 1 legend should be revised to represent the data appropriately. Data are currently presented as “n” (%) rather than OR(CI).

ASD should be defined in the table legends where necessary.

General comments

The phraseology of the result interpretation/section should be revised, as having an association does not necessarily mean causation. This applies to the conclusion section as well.

The word ‘WE’ was used severally, this is not entirely scientific.

The entire manuscript must be thoroughly revised and rid off all discrepancies.

Author Response

We addressed all the question point-by-point as a PDF attachment.

Round 2

Reviewer 1 Report

The revised manuscript merits publication

A  few comments:

Mild English editing recommended:

Examples

Lines 26-28: "Compared with Non-OSA group, male with OSA with surgery group has the aOR of infertility was 2.70 with significant difference (95% CI, 1.46–4.98, p=0.0015), however, no significance exists in female with OSA" 

line 29: the data was stratified

line 35 - should/would not wound

lines 215-216 - i assume the authors wanted to write surgery may be considered for higher severity OSA or CPAP refractory cases (not CPAP in refractory cases)

line 218 - highly is not needed in the sentence (higher severity of OSA is associated...)

Line 225 sentence beginning with That - unclear, perhaps was supposed to be a continuation of the previous sentence

Line 227 - little evidence and not few evidence 

lines 288 and 290 - wound...

Methods:

OSA was defined based on three outpatient visits or one admission.  In line 90 the authors write at least two outpatient visits.

Discussion

line 247 reference 44 - before discussing the discrepancy - need to state that the finding of the references study was that OSA is associated with female infertility.

The low incidence rate in the referenced study and the younger age group would only act to reduce that study's ability to detect an association between OSA and female infertility and yet an association was found. A possible reason for the difference is as the authors said the small number of patients in the referenced study that may induce bias. I recommend rephrasing this section both for English and content. May leave the limitation of number of participants and possibly conclude that further study is needed to better clarify the association of OSA with female infertility. Can also add the difficulties in studying such an association (such as the more complex/diverse causes of female infertility, the prevalence of OSA in women and the lag time from OSA to possibly manifest as infertility.

Author Response

Thank you for considering our manuscript for revision. All the response point by point as attachment.

Reviewer 2 Report

The authors have made a lot of corrections, and I believe the manuscript is in a better form.

However, the current version of the manuscript should be proofread to correct all typographical errors.

Few typos

Line 28- When the data were stratified according…..

Line 35- Furthermore, more studies would be encouraged….

Minor comments

Line 57-58 Whether sleep apnea is a risk factor for incidental infertility in females and males is examined in this study. Kindly rephrase the aim of the study. E.g “Therefore, the current study examined whether sleep apnea is a risk factor for incidental infertility in females and males”

Table 1 legend: values in bracket should be defined i.e values in brackets () = percentage of population size

Author Response

(The authors gave the same response as above.)
